# Cholate Disrupts Regulatory Functions of Cytochrome c Oxidase

**DOI:** 10.3390/cells10071579

**Published:** 2021-06-23

**Authors:** Rabia Ramzan, Jörg Napiwotzki, Petra Weber, Bernhard Kadenbach, Sebastian Vogt

**Affiliations:** 1Biochemical-Pharmacological Center, Cardiovascular Research Laboratory, Philipps-University Marburg, Karl-von-Frisch-Strasse 1, D-35043 Marburg, Germany; ramzan@med.uni-marburg.de (R.R.); weberpe@staff.uni-marburg.de (P.W.); 2Department of Heart Surgery, University Hospital of Giessen and Marburg, D-35043 Campus Marburg, Germany; 3Theodor-Litt-Realschule, D-40593 Düsseldorf, Germany; sekretariat.theodorlittstr-rs@schule.duesseldorf.de; 4Department of Chemistry, Philipps-University, D-35032 Marburg, Germany; kadenbach@staff.uni-marburg.de

**Keywords:** cholat, cytochrome c oxidase, ATP binding site, regulatory function, allosteric ATP-Inhibition

## Abstract

Cytochrome c oxidase (CytOx), the oxygen-accepting and rate-limiting enzyme of mitochondrial respiration, binds with 10 molecules of ADP, 7 of which are exchanged by ATP at high ATP/ADP-ratios. These bound ATP and ADP can be exchanged by cholate, which is generally used for the purification of CytOx. Many crystal structures of isolated CytOx were performed with the enzyme isolated from mitochondria using sodium cholate as a detergent. Cholate, however, dimerizes the enzyme isolated in non-ionic detergents and induces a structural change as evident from a spectral change. Consequently, it turns off the “allosteric ATP-inhibition of CytOx”, which is reversibly switched on under relaxed conditions via cAMP-dependent phosphorylation and keeps the membrane potential and ROS formation in mitochondria at low levels. This cholate effect gives an insight into the structural-functional relationship of the enzyme with respect to ATP inhibition and its role in mitochondrial respiration and energy production.

## 1. Introduction

In mitochondria cytochrome c oxidase (CytOx) represents the rate limiting step of the respiratory or electron transmission chain (ETC). The mitochondrial membrane potential established by ETC across the inner membrane is used by the ATP synthase to generate ATP from ADP and phosphate [1,2]. In different experimental settings, isolation procedures especially those involving ligands, come into consideration because of their roles in inducing conformational changes that ultimately result in different enzymatic activities. We have shown that reversible dimerization of CytOx regulates mitochondrial respiration mediated by Calcium and cAMP. This mechanism has recently been shown in intact isolated rat heart mitochondria [3] describing how mitochondrial respiration is controlled by the ATP/ADP-ratio via the “allosteric ATP-inhibition” [4]. This mechanism guarantees optimal ATP synthesis, moderate oxygen uptake and low generation of reactive oxygen species (ROS) [5] and is stated as the “second mechanism of respiratory control” [6]. Although different mechanisms regulate the function of CytOx [7,8], the inhibition of CytOx by ATP occurring at very high ATP/ADP ratios [9] is allosterically controlled by the dimeric conformation [3]. However, to work with the isolated enzyme requires the enzymatic preparation with cholate [10,11] that actually induces a stable dimerization of CytOx itself [12]. Cholate binds to the enzyme at the ATP/ADP binding sites and it is already known that the extramitochondrial ATP/ADP-ratios regulate cytochrome c oxidase activity via binding to the cytosolic domain of subunit IV [13].

The enzyme of vertebrates is composed of 13 subunits [14]. The catalytic subunits I-III are encoded by mitochondrial DNA and synthesized within mitochondria. Ten additional (supernumerary) subunits are encoded by nuclear DNA and must be translocated into mitochondria [15] and assembled into the holoenzyme [16]. The functions of nuclear-encoded subunits, which occur partly in tissue-developmental and species-specific isoforms [17,18], are still not fully known. Some regulatory effects on the catalytic activity of CytOx have been reported for subunit IV [4,19,20,21], subunit Va [22], subunit Vb [23,24], and subunit VIa [25,26].

In bovine heart CytOx, 10 high-affinity binding sites for ADP have been identified by equilibrium dialysis [27], 7 of which can be exchanged by ATP at very high ATP/ADP-ratios [9,28]. The identified nucleotide binding site at the matrix domain of subunit IV was suggested to induce the “allosteric ATP-inhibition of CytOx” when bound ADP was exchanged by ATP at high ATP/ADP ratios (half-maximal at ATP/ADP = 28) [29]. The sigmoidal inhibition curve in the kinetics of oxygen uptake against various ferrocytochrome c concentrations (Hill-coefficient = 2) indicate the cooperativity of two binding sites for cytochrome c [4].

It is suggested that the first binding site is located at each monomer while the second is found only in the dimeric enzyme. Hence, both monomers contribute to the second binding site [3]. The molecular structure of crystallized CytOx was first described in 1996 by Tsukihara et al. [30]. The enzyme was isolated from bovine heart mitochondria using sodium cholate and crystallized in the presence of dodecyl maltoside. However, it was found that the enzymatic crystals contained 10 molecules of cholate in each monomer [28], which were suggested to be located at the 10 ADP binding sites, since cholate and ADP have a similar structure. The addition of cholate to CytOx, isolated in non-ionic detergents, dimerizes the enzyme [12]. Additionally, the kinetic stability of CytOx (enzymatic activity and loss of subunits), isolated with non-ionic detergents, was significantly increased by the addition of cholate or other bile salts [31]. While in contrast to cholate, the interaction between monomers in a dimeric CytOx in non-ionic detergents is much weaker since the increasing concentrations of the detergent monomerizes the enzyme [32].

In this article, our studies describe the abolishment of the “allosteric ATP-inhibition of CytOx” by cholate, which replaces the bound ADP and thus stabilizes the dimeric enzyme [31]. By the UV/VIS absorption spectroscopy and the measurement of enzyme kinetics, we show changes in the conformation and activity. Therefore, we conclude that the crystal structure of the enzyme isolated by the use of cholate is also dimeric [31,32], but is different from the physiological structure of CytOx.

## 2. Methods

### 2.1. Isolation of the Enzyme

CytOx was isolated from bovine heart mitochondria by Triton X-114 and Triton X-100 (used for solubilization); chromatography on a DEAE-Sephacel column, followed by ammonium sulfate fractionation in the presence of 1% sodium cholate as described previously [10,33]. CytOx was isolated from mitochondria by removing the membrane and matrix proteins with nonionic detergents. Subsequently, the purification by column chromatography and fractional precipitation by ammonium sulfate, were performed. All steps were carried out at 4 °C. After the individual steps, the heme aa_3_ content of each sample was determined photometrically (difference spectrum dithionite reduced—air oxidized). Approximately 200 mL of mitochondrial suspension was thawed overnight at 4°C. Under continuous stirring, ¼ volume of KCl-solution was added and then Triton X-114 solution was added dropwise. After complete addition of the solution, stirring was continued for a further 10 min followed by centrifugation at 48,000 rpm for 30 min. The supernatant was carefully poured off and discarded. The sediment was suspended in a buffer of 200 mM KCl and 10 mM Tris-HCl (pH 7.2). Homogenization of the suspension was performed using a Potter homogenizer and centrifugation was performed again as described before. The supernatant was discarded and the pellet was resuspended in the same volume of a buffer with a pH of 7.2 containing 200 mM KCl, 10 mM Tris-HCl and 5% Triton X-100 (*w/v*). The suspension was thoroughly pottered and centrifuged again for 30 min at 48,000 rpm (130, 142.1 g).

After this centrifugation step, the supernatant predominantly contained the bc1-complex as well as small amounts of CytOx and thus was discarded. The pellet was suspended in the same volume of the same buffer, pottered, and centrifuged again. The collected supernatant contained CytOx, and the pellet was discarded (unless it still retained the significant amounts of the enzyme). The CytOx-containing supernatant was diluted four times with double distilled water. The solution was then applied to a DEAE-Sephacel ion exchanger that was freshly equilibrated with a buffer of 10 mM Tris-HCl (pH 7.2) and 0.1% Triton X-100 (*w/v*). This resulted mainly in the binding of CytOx and bc1 complexes. Subsequently, the column was washed with at least 3 column volumes of a buffer containing 10 mM Tris-HCl (pH 7.2) and 0.1% Triton X-100 (*w/v*) to remove the unbound proteins. Afterwards, performing elution with a buffer of pH 7.2 that contained 200 mM NaCl, 10 mM Tris-HCl and 0.1% Triton X-100 (*w/v*), the CytOx was collected as a green fraction. At this step, CytOx was also separated from the bc1 complexes which eluted significantly later. The CytOx-containing eluate was mixed with 1% (*w/v*) solid sodium cholate. Precipitation by ammonium sulfate (AS) was performed by adding drop by drop cold-saturated solution of Ammonium Sulfate until the concentration of 28% was reached. The pH of the solution was controlled and maintained to the value of pH 7.2–7.4 during this step. This precipitation was carried out overnight. The suspension was then centrifuged at 15,000 rpm (20,307× *g*) for 15 min and the sediment obtained was discarded. The collected green supernatant was successively subjected to several more precipitations at various ammonium sulfate concentrations of 37, 42, and 46%. These precipitations and resulting suspensions were then centrifuged. The individual pellets were finally dissolved in 250 mM sucrose and 10 mM Tris-HCl (pH 7.2) buffer, aliquoted, and were frozen at −80°C until further use. The purity of isolated CytOx was verified by recording a spectrum as well as by performing SDS-PAGE. Clean, air-oxidized CytOx had an E280 nm/E420 nm—quotient of 2.6–2.8 and showed 13 bands in SDS-PAGE. For further experiments, the 46% AS precipitated enzyme fraction was used.

### 2.2. UV/VIS Absorption Spectroscopy

The UV/VIS absorption spectra were measured with a monochromatic double beam photometer (Uvikon 240™). After recording the baseline (zero adjustment), a few grains of sodium dithionite were added, which resulted in the reduction of bc1 complex, as well as of cytochrome c and of CytOx followed by a subsequent change in the spectrum. The difference spectrum in the range 500–650 nm was recorded. The extinction coefficient of cytochrome c oxidase is ε_(605 of 630 nm)_ = 24 mM^−1^ cm^−1^.

### 2.3. Equilibrium Dialysis

Using equilibrium dialysis, the binding of nucleotides to CytOx under reversible conditions were studied. A spherical chamber was divided into two equal halves by a dialysis membrane. Both chambers were filled with 100 μL of a buffer solution (50 mM potassium phosphate, 100 mM KCl, pH 7.4 and 1% Tween 20) containing [^35^S]ATPαS (DuPont NEN (Bad Homburg, Germany) or [^35^S]ADP αS, added at various concentrations of 2.5–250 μM. Additionally, in one of the two halves, 5 μM CytOx was added. Binding of the nucleotide to the enzyme caused radioactivity to accumulate in the part of the chamber that contained the enzyme. Dialysis was performed for 72 h at 4 °C with shaking. Radioactivity was measured in a scintillation counter.

### 2.4. Kinetic Measurements

Mitochondria for kinetic measurements were isolated by standard procedures in an isolation medium (250 mM sucrose, 10 mM Hepes, 1 mM EDTA, and 0.2% fatty acid-free bovine serum albumin, pH 7.4 at 4 °C). CytOx activity was measured polarographically. Details are given in the legends to the figures. Measurements of CytOx kinetics with intact rat heart mitochondria were performed as described previously [33], i.e., at 25 °C in 0.5 mL of kinetics buffer (250 mM sucrose, 10 mM Hepes, 5 mM MgSO_4_, 5 mM KH_2_PO_4_, 0.5% fatty acid free BSA, 0.2 mM EDTA, pH 7.2) with or without 5 mM ADP or 5 mM ATP with an ATP-regenerating system. However, in these measurements, the ATP regenerating system consisted of GAP (glycerinaldehyde phosphate) + GAPDH (glycerinaldehyde phosphate dehydrogenase) + NAD^+^ (NAD), instead of PEP (phosphoenolpyruvate) + PK (pyruvate kinase) because GAP and GAPDH generate ATP in glycolysis similar to PEP and PK. The kinetics buffer contained 18 mM ascorbate (neutralized with KOH between pH 6 and 7) to maintain cytochrome *c* in the reduced form. The rate of oxygen uptake owing to the autoxidation of ascorbate was subtracted from the rates measured in the presence of mitochondria. Later, these rates were divided by the respective mitochondrial protein concentrations. Finally, using GraphPad Prism version 5.0 (GraphPad Software Inc., La Jolla, CA, USA) software, data were plotted on graphs as mean ± SEM and regression analysis was performed.

## 3. Results

Binding of cholate to the enzyme during the isolation procedure results in a dimeric form of CytOx [12]. The enzyme was isolated from bovine heart mitochondria using sodium cholate as a detergent and crystallized in the presence of dodecyl maltoside according to [28]. The spectrum of CytOx isolated with cholate is different from the spectrum of the enzyme incubated overnight with 5 mM ADP (Figure 1). To the contrary, the enzyme spectrum remained almost identical to that of the isolated cholate-CytOx after overnight incubation in the presence of 5 mM ATP. However, it was found that the exchange of cholate by ADP is a very slow process with a half-time of about 1.5 h at 0 °C, as measured by UV/VIS absorption spectroscopy.

Although the maximum spectral change between ADP-CytOx and ATP-CytOx (in the reduced forms) occurs at 100% ADP, but already at 1% ADP, more than half-maximal spectral change is obtained with the ATP-CytOx (Figure 2).

The enzymatic activity of cholate-CytOx incubated overnight in the presence of ADP is higher as compared to the cholate-CytOx (control i.e., incubated without nucleotides). While the activity of CytOx incubated overnight in the presence of ATP, it is similar to that of the cholate-isolated enzyme (Figure 3). All curves are hyperbolic in shape including the curve of ATP. Here, the lack of “allostery”, i.e., the inhibition of CytOx activity at very high ATP/ADP-ratios [9], is apparently due to the dephosphorylation of the cholate-isolated enzyme [3].

The slow exchange of cholate by ADP or ATP following 24 h dialysis in isolated cholate-CytOx is shown in the kinetics of Figure 4. When samples were incubated overnight with ATP (closed squares) or ADP (closed triangles), then cholate was removed from the binding sites of the enzyme. Subsequent measurements of enzymatic activities in these samples showed higher activity in ADP incubated cholate-CytOx and lower in case of ATP incubated cholate-CytOx. Furthermore, addition of 5 mM ADP into the ATP incubated sample directly before measurements (open squares) resulted in an instant release of the ATP inhibitory effect. Thus, ADP appears to have higher affinity to the binding sites than ATP. Moreover, ADP addition prior to activity measurement into the isolated cholate-CytOx (incubated overnight without nucleotides) revealed low activity (open triangles). Apparently, the exchange of bound ATP by ADP on CytOx occurs instantly, which may result in a conformational change due to the nucleotide binding and affinity. The exchange of cholate by ADP seems to be a slow process, because in a short time frame ADP is not able to replace cholate from the dimeric enzyme. Besides, due to the molecular similarity of ATP and cholate, the dimeric status of the enzyme appears to be ATP dependent and thus show decreased enzymatic activity. However, cholate is eventually removed from the enzyme either by ATP or ADP, depending on the time duration.

The influence of cholate on the “allosteric ATP-inhibition of CytOx” in isolated intact rat heart mitochondria is presented in Figure 5. Isolated rat heart mitochondria have a partially disrupted outer membrane, and added cytochrome c reacts with CytOx of those areas only where the inner membrane is accessible. In the absence and presence of 0.1% cholate, the oxygen consumption increased at increasing cytochrome c concentrations with and without the addition of 5 mM ADP. While the addition of ATP was produced by the ATP-regenerating system (GAPDH + GAP + NAD+), it resulted in the full inhibition of oxygen consumption almost until 100 µM cytochrome c. However, in the presence of 0.5 and 1% cholate, the inhibition of oxygen consumption was fully released even at low cytochrome c concentrations. For example, in the kinetic curves of rat heart mitochondria without any additions, the consumption of oxygen at 30 µM cytochrome c increases from 20 to 80 nmole O_2_ × min^−1^ × mL^−1^ when 1% cholate was added (see arrows in Figure 5), suggesting the removal of the outer membrane by the detergent and thus promoting the accessibility of added cytochrome c to almost all the CytOx at the inner membrane.

## 4. Discussion

### 4.1. Cholate Mimicks the Nucleotide Binding on CytOx

ATP is a nucleotide that consists of three main structures: the adenine nitrogenous base, the ribose sugar, and a chain of three phosphate groups bound to ribose. The phosphate tail of ATP is the actual power source that the cell taps. In case of cholate, the nonameric molecular ring structure equal to the adenine and the aliphatic arm with a double bound oxygen molecule (ketone form) enables its binding on CytOx at the ATP/ADP sites. Because of the different molecular charges (ADP = −3, ATP = −4), we suggest associated conformational changes and different binding affinities in the holoenzyme [7]. Although the molecular weights of ADP, ATP and cholate are slightly different (427.2 g/mol vs. 507.18 g/mol vs. 430.6 g/mol) the topological polar surface area of cholate is clearly reduced (233 Å² and 279 Å² vs. 98 Å²). This helps to understand why the cholate molecule easily enters the CytOx during the isolation, and result in structural changes at the surface as well as an altered proton pumping stoichiometry [35]. Moreover, it stabilizes the holoenzyme and supports the conformational change of the binuclear cytochrome a3-CuB-center. The first crystal structure of dimeric eukaryotic CytOx (Figure 6) was published in 1996 by Tsukihara et al. [30]. The enzymatic crystals were very stable and were found to contain 10 cholate molecules per monomer identified by using radioactive cholate [28].

Since then, numerous molecular structures of proteins including membrane bound enzyme complexes have been determined by X-ray crystallography. In order to prepare crystals of enzyme complexes of membrane proteins, their isolation is required. Cholate was found as an ideal detergent for extraction and purification. This holds true for CytOx crystals. The enzyme from bovine heart contains 10 high-affinity binding sites for ADP, from which 7 can be exchanged by ATP at high ATP/ADP ratios [27], and because of the similarity in structures, the naturally bound ADP in CytOx was assumed to be exchanged by cholate. Robinson and coworkers found that isolation of bovine heart CytOx using non-ionic detergents such as dodecyl maltoside, results in a dimeric enzyme with weak interactions, while an increase in detergent concentration monomerizes the enzyme again [32]. However, addition of sodium cholate profoundly dimerizes the enzyme [12] and significantly increases its kinetic stability [31]. In fact, Kyoko Shinzawa-Itoh never succeeded until now in preparing enough CytOx for crystallization without using cholate during purification (personal communication to B. Kadenbach).

### 4.2. Cholate Induces Dimerization and Reduces Complex IV Activity

Structural analyses of the mitochondrial respiratory supercomplexes revealed that the CytOx monomer associates with complex I and complex III of the electron transport chain (ETC), indicating that the monomeric state is functionally important [37]. In previous structural analysis of bovine CytOx [11,30], the enzyme was solubilized from mitochondrial membrane using cholate followed by ammonium sulfate fractionation in the presence of cholate, and finally the detergent was replaced by n-decyl-β-d-maltoside. Recently, monomeric and dimeric (m/d) bovine CytOx were prepared and stabilized by the same group using the amphiphilic polymer amphipol, and it was shown that the monomer had higher activity [37]. The most common supercomplex found was I_1_ III_2_ IV_1._ Additionally, they found that the monomer/dimer activity ratio is strongly pH dependent. At pH 6.0, the activity ratio was almost 2.5 to 1 (m/d), whereas at more physiological pH of 7.4, the ratio was 4 to 1 (m/d) with reduced total enzymatic activity. The data support the idea of an I_1_ III_2_ IV_1_ composition of the ETC under physiological condition and supercomplex conversion to higher dimeric content in case of stress. It supports Kadenbach’s theory postulating that allosteric inhibition needs a dimeric structure of CytOx, in order to limit the enzymatic activity and to avoid excessive formation of reactive oxygen species (ROS) [38]. It is a new mechanism which functions independently of the Mitchell’s Theory and keeps Δψ_(m)_ at low values through feedback inhibition of complex IV at high ATP/ADP ratios, thus maintaining high efficiency of oxidative phosphorylation. In the present study, cholate mimics the nucleotid-binding to the enzyme.

The spectrum of isolated CytOx with bound cholate differs to that of CytOx bound with ADP (Figure 1 and Figure 2). The kinetic measurements of the samples where nucleotides and cholate were exchanged following overnight incubation show that the enzymatic activities in the presence of ADP on one hand and ATP or cholate on the other hand differ (Figure 3) Despite hyperbolic kinetics in all cases, activities with ADP were higher while with that of ATP or cholate were lower. In case of the latter ones, both curves show a similar course. In another experiment (Figure 4), when the kinetics of these overnight incubated samples were measured in the presence of ADP added additionally just before the measurement, similar results were found with ADP, i.e., higher hyperbolic curve. Interestingly, the addition of 5 mM ADP directly before activity measurement to the ATP incubated sample resulted in an instant release of the inhibitory effect, and the activity went higher, almost to the rate of ADP. However, in the sample incubated overnight without nucleotides, followed by the enzymatic activity measurement by adding ADP prior to measurement did not increase the oxygen consumption and thus the activity remained inhibited, indicating the importance of time duration required for the exchange of cholate by ADP.

### 4.3. Different Binding Properties of Nucleotids and Cholate Results in Different Enzymatic Activities of CytOx

When cholate-CytOx is incubated with ADP or ATP, it results in dissociation of cholate due to decreased affinity as compared to nucleotides (cholate < ADP or ATP). However, an immediate addition of ADP to the ATP incubated sample results in higher activities than that of ATP due to its greater affinity (ADP > ATP). Furthermore, the addition of ADP to the cholate CytOx directly before activity measurement does not increase the enzyme activity to the normal ADP level. This may be due to the slow exchange of bound cholate by ADP thus depending on the time duration. These results indicate that the detection of conformational change by UV/VIS spectroscopy is not only related to the different enzymatic activities due to different binding properties of ATP, ADP and cholate, but also that some processes are time dependent. Nonetheless, all these factors influence the activity of mitochondrial respiration, and this represents the core of the “second mechanism of respiratory control” [4,6].

From the spectral change of CytOx after binding of cholate and from its slow exchange by ADP, we conclude that the X-ray structures or cryo-EM structures of CytOx, isolated by the use of cholate, do not fully represent the physiological structure of CytOx in cells. Furthermore, this is in relevance to the recent discussions about the number of protein subunits in the monomeric enzyme [39].

### 4.4. Functions of CytOx Subunits and Supercomplexes Are not Fully Understood

Since cholate can induce the dimerization of CytOx by occupying the ADP/ATP binding sites, so in addition to the already found phosphorylations and acetylations of the dimeric CytOx [40], the question remains if the cholate-isolated dimer is identical to the physiologically existing dimeric structure of the enzyme, and finally the role of NDUFA4 associated with the holoenzyme needs to be clarified. Indeed, in a new module-based assembly of CytOx, the addition of NDUFA4 enables dimerization [41]. Nevertheless, the most common supercomplex found is composed of I_1_III_2_ IV_1_ [37]. Moreover, if NDUFA4 would represent as an essential subunit of CytOx, it would have also been found in the dimeric structure of CytOx reported by Tsukihara et al. [30]. The establishment of feedback inhibition of CytOx by ATP, which may occur due to the cooperativity between two binding sites of cytochrome c in the dimeric enzyme (the allosteric ATP-inhibition of CytOx), is suggested to be impossible in monomeric NDUFA4-CytOx complex. Using isolated intact rat heart mitochondria, it has already been described that cAMP-dependent phosphorylation at the intermembrane side of CytOx subunit I induces a dimeric structure of the enzyme that may be responsible for establishing the allosteric ATP-inhibition [14,42,43]. Recently, Hartley et al. presented the cryo-EM structure of *S*. *cerevisiea* Complex IV in a III_2_IV_2_ supercomplex where they did not find the NDUFA4 homologue [44]. An interaction between COX7A and Complex I subunit NDUFB8 was also observed in the porcine respirasome [16]. These points remain to be clarified as well as further studies on the molecular composition of cholate-CytOx-dimer are needed. Correspondingly, the functions of known phosphorylation sites [45,46,47] as well as the identification of those sites responsible for inducing dimerization or strings of ETC are still not known [48].

### 4.5. Cholate May Hamper the Formation of Mitochondrial Supercomplexes

Although dimerization stabilizes the CytOx-holoenzyme [31] but in the most commonly identified supercomplex, the enzyme appears as a monomer with higher activity [37], this represents a paradox. In Figure 5, we present the data of polarographic measurements of CytOx kinetics at different cholate concentrations. Since cholate is a lipophilic bile acid which can easily penetrate the mitochondrial membranes, it can thus act on all the elements of the ETC. Consequently, with an increased cholate concentration, the mitochondrial oxygen consumption accelerates and the allosteric ATP-inhibition is released, although the enzyme remains dimeric. Apparently, this seems contradictory to our hypothesis, but it can be better understood with a change in the molecular structure of CytOx by cholate or nucleotides. Shoji et al. found modifications in the binding affinity of the oxidized binuclear center in the presence of cholate or nucleotides and concluded that the structural relationship between the known cholate-binding site and the binuclear cytochrome a3-CuB site influences the electron transmission [49]. Indeed, Rolo et al. found that at increased concentrations of bile acids, a decrease in the mitochondrial membrane potential (Δψ_m_) and state 3 respiration appears, but the state 4 respiration as well as the acidification, both increases [50]. Consequently, the respiratory control ratio (RCR), which represents the maximum increase in mitochondrial oxygen consumption in isolated mitochondria and can be achieved above the LEAK oxygen requirement when driving the phosphorylation of ADP into ATP, is decreased. This could also be the reason of bile acid toxicity on organic tissues [51,52,53].

The formation of respiratory supercomplexes may have advantages, because the latest data indicate that electron transfer between complexes III and IV represents a rate limiting step. Hence, incorporating these enzyme complexes in the mitochondrial membrane to form supercomplexes may improve the rate of electron transmission [54] while failure to form these supercomplexes may results in the impaired rates. In the control sample of Figure 5, we found an allosteric ATP inhibition because of the ATP regenerating system (GAPDH, GAP and NAD). As stated earlier, cholate binds closer to the binuclear cytochrome a3-CuB site [49] and it has also been reported that cholate readily displaces almost all of the native phospholipids in the molecule [55], therefore, the supercomplexes are not formed [56]. Consequently, the ETC gets a new order and/or the complexes fall back into their singularities. This may be due to the reason when only one of their component complexes is missing, then supercomplexes are not formed. Relevantly, a delay is known between the formation of the individual complexes and that of the supercomplexes [57]. Hence, cholate not only disrupts the original supercomplex formation of the ETC but may also hinder the subsequent electron transfer rate as shown in different kinetic measurements of the enzymatic activity. Moreover, an influence of a pH shift cannot be neglected because it also effects the dimeric/monomeric ratio [37,58]. Interestingly, our experimental findings of an increase in specific activity of CytOx at 0.1 and 1% cholate are in accordance with the studies of late 1970s [59]. Notably, recent studies imply that binding of physiological ligands that are structurally similar to cholate could trigger dimerization in the mitochondrial membrane, and that the non-specifically bound lipid molecules at the transmembrane surface between monomers support the stabilization of the dimer. The weak interactions involving the transmembrane helices and extramembrane regions may play a role in positioning each monomer at the correct orientation in the dimer.

A reversible monomer/dimer transition may contribute to the enzymatic regulation [60]. Changing the supercomplex composition could reflect nothing else but the availability and tissue specific demand of oxygen [61]. The loss of supercomplex organization also in terms of oxidative stress results in an inefficient electron transfer and decreased oxidative phosphorylation. This may be significant in case of pathophysiological findings such as the toxic effects of bile acids [62] or emergence of atrial fibrillation [63].

## Figures and Tables

**Figure 1 cells-10-01579-f001:**
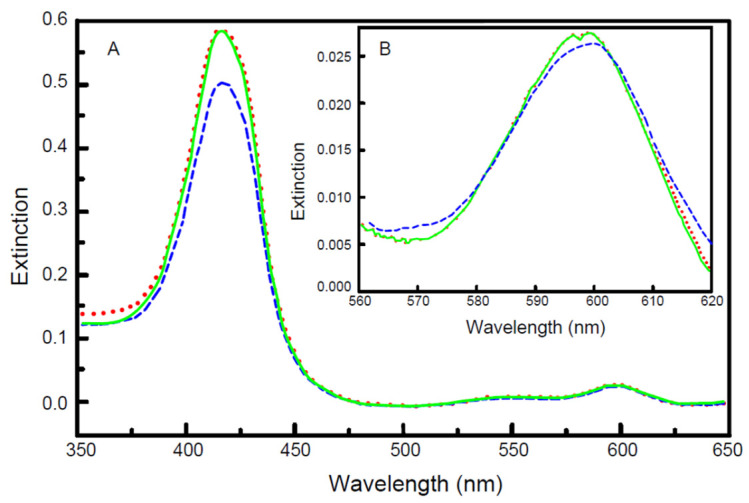
Absolute UV-VIS absorption spectra (maximum at 421 nm) of isolated cholate-bound CytOx from bovine heart. The 3 µM CytOx concentration was dissolved in 50 mM potassium phosphate, 1 mM EDTA, pH 7.4 and 1% Tween 20. Samples were incubated for 24 h at 0 °C with 5 mM ATP, or 5 mM ADP, or without nucleotides. The measured absorption spectrum without nucleotides is a continuous line in green while in the presence of ATP is the dotted line in red and that of ADP is a broken line in blue.

**Figure 2 cells-10-01579-f002:**
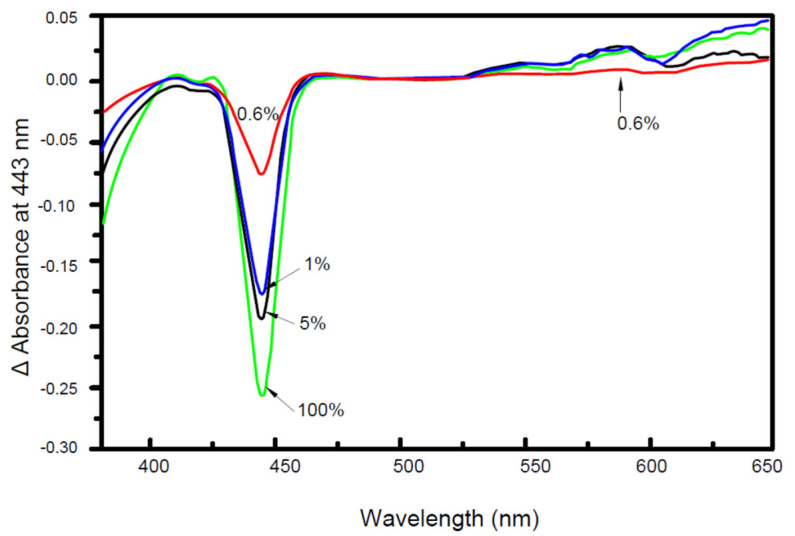
Spectral changes of the reduced form of isolated cholate-CytOx from bovine heart by ADP or ATP. The UV-VIS absorption difference spectra were measured with 100% ATP (5 mM) in the reference cuvette. While the sample cuvette contained the indicated percentage of ADP i.e., 0.6 to 100% (ADP + ATP = 5 mM = 100%). Before measurements, samples containing 3 µM of isolated CytOx in 50 mM potassium phosphate (pH 7.4), 1 mM EDTA, 1% Tween-20 and the indicated nucleotides were incubated for 24 h at 0 °C. Just before measurement, 2 mM sodium dithionite was added for reduction. The optical spectrum of heme a is red-shifted as known in case of aa_3_-type cytochrome c oxidases. Early spectroscopic studies in *Paracoccus denitrificans* indicated that this may be due to hydrogen-bonding of the formyl group of heme a to an amino acid in the close vicinity. Even here, the optical spectral shift of native heme a is suggested because of a hydrogen-bonding interaction between the formyl group and arginine-54 in subunit I of cytochrome aa_3_, and that a smaller part is due to an electrostatic interaction between the D ring propionate of heme a and arginine-474 [34].

**Figure 3 cells-10-01579-f003:**
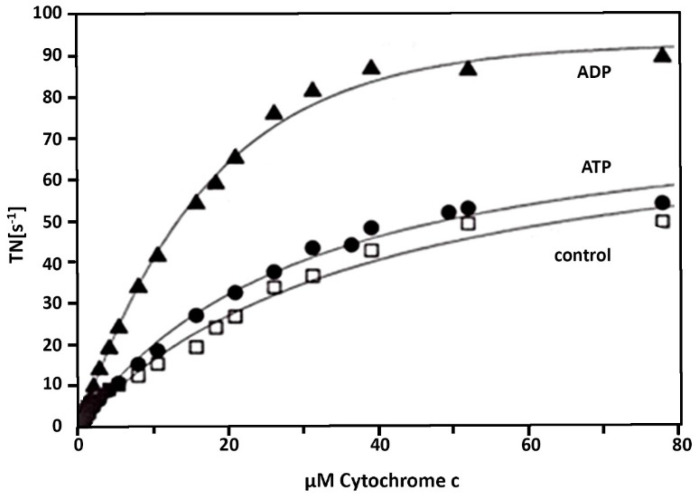
Activity of isolated cholate-CytOx from bovine heart in the presence of ADP, ATP, or without nucleotides after incubation of 0.1 µM CytOx for 24 h at 0 °C in 50 mM potassium phosphate (pH 7.4), 1 mM EDTA and 1% Tween 20 in the presence of 5 mM ATP (●), 5 mM ADP (▲) or in the absence of nucleotides (control □). Oxygen uptake was measured at increasing cytochrome c concentrations. TN = turnover number (mole O_2_ × s^−1^/mole heme aa_3_). Note the resemblance in the course of activity between control vs. ATP bound to cholate-CytOx.

**Figure 4 cells-10-01579-f004:**
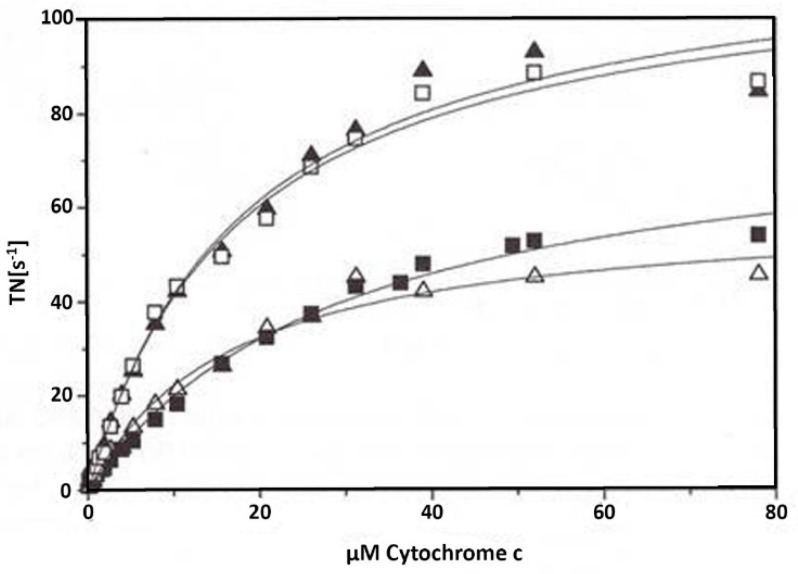
Dependence of cholate-CytOx activity on the time of incubation in the presence of nucleotides: The 0.1 µM concentration of bovine heart CytOx was incubated for 24 h at 0 °C in 50 mM potassium phosphate (pH 7.4), 1 mM EDTA, and 1% Tween 20. Incubation with 5 mM ATP (■) or 5 mM ADP (▲) results in lower or higher activities, respectively. Furthermore, just before activity measurements, the addition of 5 mM ADP into the ATP incubated sample results instantly in the removal of the inhibitory effect (□). Contrarily, the addition of 5 mM ADP into the cholate-CytOx sample just before measurement (∆) shows that the enzymatic activity remains low, which is comparable to the activity measured in the presence of ATP. Here it is obvious that ADP cannot immediately replace the cholate at the binding sites of the enzyme. TN = turnover number (mole O_2_ s^−1^/mole heme aa_3_).

**Figure 5 cells-10-01579-f005:**
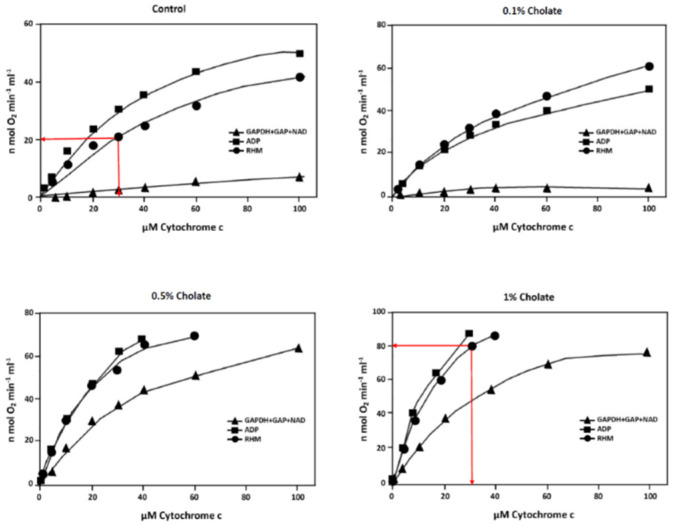
Effects of different cholate concentrations on the activity and allosteric ATP-inhibition of CytOx in isolated intact rat heart mitochondria (RHM). Mitochondria (30 µg/mL) were titrated with increasing concentrations of cytochrome c in a medium containing 250 mM sucrose, 10 mM Hepes, 5 mM MgSO_4_, 5 mM potassium phosphate, 0.5% BSA (fatty acid-free), 0.2 mM EDTA, pH 7.2. Where indicated, 5 mM ADP was added. High ATP/ADP ratios were established by the addition of 5 mM ATP and the ATP-regenerating system: 1.2 mM GAP + 100 units/mL GAPDH + 1 mM NAD^+^ (GAPDH + GAP + NAD). In comparison to control, the oxygen consumption at 30 µM cytochrome c increases from 20 to 80 nmole O_2_ x min^−1^ x ml^−1^ in the presence of 1% cholate without any additions (red arrows).

**Figure 6 cells-10-01579-f006:**
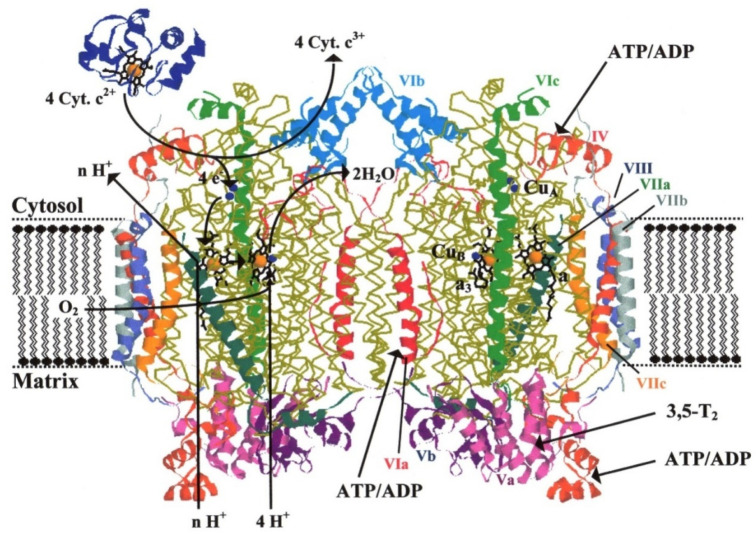
Crystal structure of dimeric bovine heart CytOx [26]. Coordinates (PDB entry 1occ)) were processed with the Swiss PDB Viewer/Pov Rayprogram (E. Schwan, Ray Tracing for the Macintosh CD, Waite Group Press, Corte Madera (CA, USA), 1994). The three mitochondrially coded subunits in each monomer are represented as peptide backbone traces (yellow) with their redox centers highlighted, while the helices of nuclear-coded subunits are numbered and labeled as color-coded ribbons in the right-hand monomer. The monomer on the left depicts schematically the pathway of electrons entering from the donor, cytochrome c (top left drawn in blue; shown in a somewhat arbitrary position), oxygen binding to the binuclear center, and pathways and stoichiometry for matrix proton uptake and release on the cytoplasmically oriented side of the mitochondrial inner membrane (with the positioning of the phospholipid bilayer solely for illustration). In the monomer on the right, the location of the potential binding sites for ATP (or ADP) and 3,5-diiodothyronine, are indicated [34] and suggested proton channels are shown at the monomer on the left side [36].

## Data Availability

Data sharing is not applicable to this article.

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
