# Peer review of "Cholate Disrupts Regulatory Functions of Cytochrome c Oxidase"

_cells, 2021, doi:10.3390/cells10071579_

Round 1

Reviewer 1 Report

The article tries to make the case that bile acids, when accumulating in the liver during cholestasis, may negatively affect mitochondrial respiration by affecting cIV. The mechanism is supposed to be mediated by direct binding of choline or the chemically similar bile acids to cIV. This would abolish binding of ADP or ATP and favour a dimeric structure of the complex. The experiments shown do not significantly extend what is known from previous studies.

With one exception, the experimental results are in vitro studies with the isolated enzyme treated or untreated with different concentrations and combinations of ADP, ATP or cholate. The experiment with isolated mitochondria does not convincingly support the conclusions.

The connection to disease is a bit far fetched and documentation is poor; some ‘impairment of mitochondrial functions’ is observed in more or less all disease conditions and the connection to bile acids as causative is too weakly documented.

Altogether, the manuscript is not clearly focused, lacks novelty, and is not well written. Below some specific points.

Abstract:

Unclear distinction between literature knowledge and results from the present study.

Introduction

Page 1, 2nd sentence: ‘…. cytochrome c oxidase (CytOx) represents the final oxygen accepting and activity controlling step of the respiratory chain….’ A little nebulous statement. What is meant by ‘oxygen accepting’? Oxygen is the final acceptor of the electrons shuffled through the electron transport chain.

These two sentences appear to be contradictive: is ADP of 7 or 10 binding sites exchanged by ATP at high ATP/ADP ratios?

Last sentence of 1st page: In CytOx from bovine heart 10 high-affinity binding sites for ADP were identified by equilibrium dialysis [23], seven of which are exchanged by ATP at high ATP/ADP-ratios [24].

Last sentence of middle paragraph on page 2: CytOx contains 10 binding sites for ADP, ten of which are exchanged by ATP at high ATP/ADP-ratios [10]. Exchange of ADP by ATP at the matrix domain of subunit IV induces the “allosteric ATP-inhibition” [9], also named “second mechanism of respiratory control” [10].

Unclear sentence (1st paragraph, 2nd page:

The „allosteric ATP-inhibition of CytOx“ was recently demonstrated in intact isolated bovine heart mitochondria and shown to be switched on by cAMP-dependent phosphorylation and switched off by Ca2+-activated dephosphorylation of CytOx [11]. What is allosteric ATP inhibition and how does it relate to regulation by phosphorylation?

Methods:

Very minimalistic descrition.

Results:

The results section is quite short, difficult to understand and lacks clear conclusions from the data shown and rationale.

Discussion

There is evidence that the monomeric cIV complex forms supercomplexes whereas dimeric cIV does not. See for example Vidoni et al. (2017). Cell Rep 18, 1727-1738. This aspect and its relationship to respiratory chain efficiency is not mentioned or discussed.

The last paragraph is highly speculative and does only marginally relate to the experiments presented in this manuscript.

Reviewer 2 Report

The subject of the paper is appropriate for the publication in the journal Cells. The submitted paper is a short manuscript dealing with the effect of cholate on cytochrome c oxidase (CytOx) activity and allosteric ATP inhibition of CytOx. However, the authors provide only limited number of experiments and some of their conclusions are far reaching without any experimental prove.   

The following issues should be clarified before publication:

Abstract:

The authors claim that the cholate induces a change of CytOx structure indicated by spectral change. However, they do not provide any experimental evidence for this statement. The last sentence of the abstract should be removed.

Introduction:

There exists redundancy in the Introduction part, e.g. the fact that “…ten high affinity sites for ADP were identified by equilibrium dialysis, seven of which are exchanged by ATP at high ATP/ADP ratios. The nucleotide binding site at the matrix domain of subunit IV was identified to induce an allosteric ATP inhibition of CytOx….” is mentioned twice in the text.

What is the meaning of the term ”kinetic stability of CytOx”?

The paragraph starting with the sentence: “The exchange of the nucleotides by cholate…” is excessively far reaching. It evokes that the interaction of cholate with CytOx should explain the mechanisms of many diseases.

Also the notion in the sentence: “Therefore, we conclude…….. and can cause liver disease” is not adequate.

Methods:

The terms spectra, spectroscopy should be changed for UV-Vis absorption spectra, UV-Vis absorption spectroscopy (this change should be done through all the manuscript).

Experiments:

Fig. 2 shows UV-Vis absorption spectra of isolated CytOx without nucleotides and in the presence of 5 mM ADP and ATP. The value of the wavelength in the maximum should be indicated in the text or in the figure. The decrease of the absorbance maximum in the spectrum of CytOx/ADP comparing to the spectra of isolated CytOx and CytOx/ATP is ca. 10%. What is the reason for this decrease? The isolated CytOx represents “fast” or “slow” form of CytOx?

The decrease of the intensity of the Soret band in the reduced CytOx in the presence of ADP is really very significant. Can you explain this observation? The label of the y- axis should be changed to “Absorbance” (Fig. 3).

All kinetics representing oxygen uptake vs. cyt c concentration show hyperbolic characteristics (Figs. 4, 5 and 6). The authors claim that: ”These data indicate the release of the allosteric ATP inhibition in CytOx by cholate”. To prove this statement, it would be helpful to experimentally show that the ATP allosteric inhibition of CytOx exists in the absence of cholate.

Discussion:

As it was already mentioned, several sentences contain excessively far reaching conclusions. In the Discussion, the sentence: “….cholate interaction with the enzyme may help to better understand liver disease due to choleastasis” is an example of these excessively far reaching conclusions. Also, the final paragraph of the Discussion (starting as “Mitochondria adapt to lipid…”) is too speculative, not based on experimental data.

Several grammatical, stylistic, and formal errors are present in the manuscript.

In conclusion, I recommend publish this manuscript as a letter, however, after major revision.

Round 2

Reviewer 1 Report

The revised manuscript is slightly improved. The novelty of this manuscript lies in the documentation of the effect of choline on CytOx, both it’s oligomeric structure and the accessibility/occupation of nucleotide binding sites.  The preparation conditions thus impact activity. The introduction presents this issue in a very unclear and incoherent way which make it very difficult to follow the arguments and the definition of the research question addressed. Given the complexity of the topic with various layers of regulatory mechanisms, the presentation of existing knowledge and research question can be very much improved. In the present form, readers who are not themselves working with CytOx and other respiratory chain complexes will be discouraged to appreciate the interesting implications of the findings.

This manuscript needs major revision.

Introduction:

‘At present, the molecular structure of the holoenzyme remains in debate.’ References to conflicting publications or publications discussing the disagreements would be helpful.

Materials and methods:

Centrifugation at 36,000 rpm; the g value should be given

Half time -> half life

Results:

Figure 1: The tracings are difficult to distinguish as they lie on top of each other. The reduction in the 420nm peak for the ADP form and the slight ‘red shift’ are not well described and interpreted. The slow cholate to ADP exchange is not documented by data, data not shown or reference to publication.

Figure 2 shows the difference spectra distinguishing the ATP from the ADP forms with the result that already small proportions of ADP result in a large fraction of the ADP form. Does this mean that the ATP form in vivo is a tiny species implicating that it does not play a role? The physiological ATP/ADP ratio is 10-20.

Figure 3 shows the activities of the cholate, ATP and ADP forms dependent on cytochrome c concentration. What is the implication of throwing the sentence ‘The lack of „allosteric ATP-inhibition of CytOx“, i.e. inhibition of CytOx activity at very high ATP/ADP-ratios [9], is apparently due to dephosphorylation of the cholate-isolated enzyme [3]’ ?

Figure 4 relates to the slow exchange of cholate for ADP. It only compares 24 h to instantaneous changes. What about 30 minutes, 2 hours etc.? The concept of ‘a conformation dependent change in affinity‘ is unclear. Explaining the different symbols with a figure legend instead of the text in the legend would be helpful for the reader.

Figure 5 attempts to connect the proposed hypothesis to an in vivo situation. Does this mean that concentrations of bile acids may affect CytOx activity in vivo? The detrimental effect of cholate on the mitochondrial outer membrane which overlays the effect on activity makes this setup very artificial. ‘RHM’ has to be explained.

Discussion

The discussion is much too long and unfocused. It is in sharp contrast to the laconic results section which very scarcely describes the experimental results. For example the monomer dimer issue of CytOx is not addressed experimentally in any way in the present work and should therefore not be discussed broadly. It should relate to the question whether purification conditions for membrane complexes affect the measurements and structures, which implications this has for such research, and how this has to be dealt with. There may also be some discussion of the in vivo relevance; could physiological cholate concentrations or concentrations of similar compounds affect CytOx activity and how big can this effect be?

‘Control measurement in Figure 6’: Figure 6 shows the structure.

Reviewer 2 Report

The authors have satisfactorily answered all questions and comments. The quality of the manuscript has been improved, especially I appreciate thoroughly elaborated discussion. Although there are still present some few grammatical, stylistic, and formal errors in the manuscript (I hope that final critical reading will eliminate them), I can fully recommend the manuscript for the publication in journal Cells.

Round 3

Reviewer 1 Report

The authors have appropriately responded to the majority of the questions and queries.

My language in the review has in parts apparently been too colloquial so that some of the sentences could be understood as 'pointed remarks'. I excuse for giving this impression; it was not my intention and I will in the future try to stick to more formal and neutral language.

Anyway, the discussion of the issues was in my impression productive and I hope that the article will get many readers including those that have not been aware of this very interesting twist of respiratory chain regulation.

 Specific comment: 

The last sentence of the introduction: 'Therefore we conclude that the crystal structure of the enzyme isolated by the use of cholate is although dimeric [31, 32], but is different from the physiological structure of CytOx.'

'Also' instead of 'although'?